# Improving genetic risk modeling of dementia from real-world data in underrepresented populations
Mingzhou Fu [1,2], Leopoldo Valiente-Banuet[1], Satpal S. Wadhwa[1], Bogdan Pasaniuc [3], Keith Vossel[1] & Timothy S. Chang [1] ✉

Genetic risk modeling for dementia offers significant benefits, but studies based on real-world data, particularly for underrepresented populations, are limited. We employ an Elastic Net model for dementia risk prediction using single-nucleotide polymorphisms prioritized by functional genomic data from multiple neurodegenerative disease genome-wide association studies. We compare this model with *APOE* and polygenic risk score models across genetic ancestry groups (Hispanic Latino American sample: 610 patients with 126 cases; African American sample: 440 patients with 84 cases; East Asian American sample: 673 patients with 75 cases), using electronic health records from UCLA Health for discovery and the All of Us cohort for validation. Our model significantly outperforms other models across multiple ancestries, improving the area-under-precision-recall curve by 31–84% (Wilcoxon signed-rank test *p*-value <0.05) and the area-under-the-receiver-operating characteristic by 11–17% (DeLong test *p*-value <0.05) compared to the *APOE* and the polygenic risk score models. We identify shared and ancestry-specific risk genes and biological pathways, reinforcing and adding to existing knowledge. Our study highlights the benefits of integrating functional mapping, multiple neurodegenerative diseases, and machine learning for genetic risk models in diverse populations. Our findings hold potential for refining precision medicine strategies in dementia diagnosis.

Dementia is a progressive syndrome marked by cognitive decline beyond what is expected from normal aging[1]. Globally, it affects about 36 million people and incurs costs of approximately $594 billion annually[2,3]. The primary etiologies of dementia include Alzheimer's disease (AD), vascular dementia, Lewy body dementia (LBD), Frontotemporal dementia (FTD), and Parkinson's disease dementia (PDD), among others[4]. Genetic predisposition plays a significant role in dementia, with numerous significant variants identified through Genome-Wide Association Studies (GWASs). For example, the Apolipoprotein E (*APOE)* gene, which encodes a protein responsible for binding and transporting low-density lipids, significantly influences the risk of late-onset AD, the most prevalent form of dementia[5,6].

Polygenic risk scores (PRSs) aggregate the effects of multiple genetic variants to quantify an individual's genetic predisposition for complex diseases like dementia[7]. A growing number of studies have underscored the robust links between AD PRS and dementia related phenotypes in the non-Hispanic white populations[8–11]. However, further research is needed to refine personal dementia genetic risk models and understand their potential limitations.

PRS performance is suboptimal in non-European ancestries, as weights for single nucleotide polymorphisms (SNPs) are mostly derived from European ancestry GWASs, limiting their generalizability[12–15]. Including causal variants like *APOE* in risk models due to their independent risk contribution is recommended, while non-causal variants can introduce noise[16,17]. Moreover, few genetic studies on dementia have been conducted within the context of Electronic Health Records (EHRs), and have predominantly focused on AD[9,18]. While AD accounts for a significant portion, many dementia cases display mixed pathologies[19,20], with mixed dementia being a common occurrence in real-world scenarios[21]. Addressing all-cause dementia could better reflect the clinical landscape and lead to advances in precision medicine that benefit a larger cohort[22].

Dementia remains significantly underdiagnosed in real-world community settings[23–28]. Early detection through genetic modeling can help healthcare providers improve diagnosis, manage symptoms effectively, and

[1]Department of Neurology, David Geffen School of Medicine, University of California, Los Angeles, Los Angeles, CA, 90095, USA. [2]Medical Informatics Home Area, Department of Bioinformatics, University of California, Los Angeles, Los Angeles, CA, 90024, USA. [3]Department of Computational Medicine, David Geffen School of Medicine at UCLA, Los Angeles, CA, 90095, USA. ✉e-mail: timothychang@mednet.ucla.edu

initiate appropriate treatments. The need for more refined methodologies to develop accurate genetic risk models across diverse populations is imperative.

In the present study, we hypothesized that the risk SNPs associated with dementia and their corresponding weights vary across diverse populations, specifically Amerindian, African, and East Asian genetic ancestries. We further proposed that the predictive performance for dementia phenotypes in non-European populations could be enhanced by identifying biologically meaningful SNPs and applying sparse machine learning models tailored to each genetic ancestry group. Thus, we present a novel approach for assessing individual dementia genetic risks across diverse populations.

To address previous limitations, we implemented several innovative measures. Firstly, we prioritized SNPs using functional and biological information based on GWAS results, focusing on causal SNPs most likely to contribute to dementia risk. Secondly, we utilized machine learning algorithms to select significant genetic variants, allowing us to fine-tune models for different ancestry groups. This method provides a notable advantage for non-European populations, which are often underrepresented in GWAS studies. Finally, we developed and validated our models within real-world EHR settings, targeting dementia as a comprehensive condition. This innovative approach holds promise for improving our understanding of individual dementia genetic risks for dementia and promoting health equity in genetic research.

## Results
### Sample description
The primary dataset for model development was derived from EHR linked to the biobank of the UCLA Health System[29]. Fig. 1 illustrates the finalized UCLA ATLAS samples, stratified by Genetic Inferred Ancestry (GIA) groups. The Hispanic Latino American (HLA) sample included 610 patients with 126 dementia cases, while the AA sample consisted of 440 patients with 84 dementia cases. The distribution of International Classification of Diseases, 10th Revision (ICD-10) diagnosis codes was relatively consistent across the two GIA samples, with Alzheimer's disease (G30) and unspecified dementia (F03) being the most prevalent diagnoses. However, the African American (AA) group exhibited a higher proportion of vascular dementia (F01) diagnoses compared to the HLA group. The East Asian American (EAA) group, with a limited case count ($N = 75$), was excluded from primary analyses but included in sensitivity analyses.

Within each GIA group, eligible controls, due to the stringent inclusion criteria, had longer spans of records and more encounters. There were no significant differences in other EHR features between dementia cases and controls (Table 1).

### Performance comparison for dementia phenotype prediction task
We developed and evaluated machine learning models to predict the binary dementia phenotype within the UCLA ATLAS sample, stratified by GIA groups. After accounting for age, sex, and ancestry-specific genetic variations (represented by principal components (PCs)), we constructed genetic risk models for dementia, incorporating offset corrections within a linearized framework. The predictive capabilities of these models were assessed using four distinct sets of genetic markers: (1) *APOE-ε4* counts, (2) AD PRS, (3) a composite of multiple PRSs, and (4) select SNPs refined through Elastic Net regularization[30]. For SNP selection, we utilized the Functional Mapping and Annotation of Genome-Wide Association Studies (FUMA) tool[31] to prioritize independent genome-wide-significant SNPs or independent gene-annotated SNPs.

Table 2 presents the overall performance of models for predicting dementia phenotypes. No discernible differences were observed among *APOE-ε4* and all PRS models (AD-PRS and Multi-PRS), regardless of the SNP set used for PRS construction—whether derived from ancestry-specific GWASs, genome-wide-significant SNPs, or gene-annotated SNPs. Notably, the predictive performance in the AA GIA sample of all PRS models was

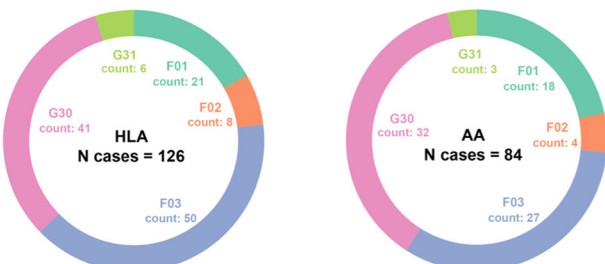

**Fig. 1 | Dementia patient characteristics by genetic inferred ancestry groups, UCLA ATLAS sample.** Distribution of diagnosis in ICD-10 codes by genetic inferred ancestry groups. AA African Americans, HLA Hispanic Latino Americans. ICD-10 codes descriptions: G30, Alzheimer's disease; F03, Unspecified dementia; F02, Dementia in other diseases classified elsewhere; F01, Vascular dementia; G31, Other degenerative diseases of nervous system, not elsewhere classified.

inferior to *APOE-ε4*, particularly evident in the Area Under the Precision-Recall Curve (AUPRC).

Elastic Net SNP models demonstrated overall improvement in dementia prediction across both GIA groups. The model incorporating gene-annotated SNPs from AD and other dementia-related disease GWASs (SNPs from AD + Neuro GWASs) emerged as the most effective, indicating a collective contribution from SNPs associated with various dementia-related diseases. Specifically, the leading Elastic Net SNP model for HLA GIA sample significantly enhanced the AUPRC by 31% (0.41 [95%CI: 0.27, 0.52] vs. 0.31 [95%CI: 0.21, 0.41], Wilcoxon signed-rank test *p*-value < 0.001), and the area under the receiver operating characteristic (AUROC) by 12% (0.73 [95%CI: 0.66, 0.79] vs. 0.65 [95%CI: 0.56, 0.72], DeLong test *p*-value = 0.01) compared to the best PRS model. Furthermore, this model outperformed the *APOE-ε4* count model, with an increase of 33% in AUPRC (0.41 vs. 0.31 [95%CI: 0.21, 0.41], Wilcoxon test *p*-value < 0.001) and 12% in AUROC (0.73 vs. 0.65 [95%CI: 0.56, 0.71], DeLong test *p*-value = 0.01).

This model's efficacy was even more pronounced within the AA GIA sample, with an increase in AUPRC by 84% (0.45 [95%CI: 0.31, 0.58] vs. 0.24 [95%CI: 0.11, 0.41], Wilcoxon test *p*-value < 0.001) and the AUROC by 16% (0.71 [95%CI: 0.63, 0.78] vs. 0.60 [95%CI: 0.46, 0.74], DeLong test *p*-value = 0.004) in comparison to the best PRS model. Relative to the *APOE-ε4* count model, the improvements were 65% in AUPRC (0.45 vs. 0.27 [95%CI: 0.16, 0.44], Wilcoxon test *p*-value < 0.001) and 17% in AUROC (0.71 vs. 0.61 [95%CI: 0.50, 0.74], DeLong test *p*-value = 0.01).

We also noted a substantial enhancement in the other performance metrics (based on the threshold that maximized the Matthews Correlation Coefficient (MCC)) of the Elastic Net SNPs models compared to other models across both GIA samples. This was evidenced by marked improvements in accuracy, precision, and the F1 score. In our sensitivity analysis, other non-linear models using gene-annotated SNPs from AD and other dementia-related disease GWASs, including Gradient Boosting Machine (GBM) and XGBoost, did not perform as well as the linear Elastic Net SNP models. Results from bootstrapping showed similar trends in model performances, as shown in Supplementary Fig. 1. Applying a more stringent $r^2$ cut-off (<0.1) for defining independent genome-wide-significant SNPs yielded results consistent with our initial findings, as detailed in Supplementary Table 1.

In summary, models leveraging SNPs as features identified through machine learning methods possess the potential to surpass those relying solely on summary scores such as PRSs in HLA and AA GIA. Furthermore, selecting SNPs mapped to genes using functional genomic data holds promise for further refining predictive performance.

### Featured risk variants and mapped genes
In our analysis of the best-performing Elastic Net SNPs models, we examined the features selected by each model. According to results from

**Table 1 | Descriptive statistics of demographic and electronic health record features by case/control groups, UCLA ATLAS sample, stratified by genetic inferred ancestry group**

| | Hispanic Latino Americans (N = 610) | | | African Americans (N = 440) | | |
|---|---|---|---|---|---|---|
| | Cases | Controls | *P*-value | Cases | Controls | *P*-value |
| N | 126 | 484 | – | 84 | 356 | – |
| Age | 78.4 (71.3, 81.7) | 75.3 (72.6, 79.6) | 0.2 | 78.0 (70.1, 82.6) | 75.7 (72.7, 79.9) | 0.7 |
| Sex (Female) | 72 (57%) | 300 (62%) | 0.30 | 46 (55%) | 218 (61%) | 0.30 |
| Span of records (in yrs) | 5.9 (2.8, 8.8) | 9.6 (7.7, 10.9) | <0.001ᵃ | 6.2 (3.1, 10.1) | 9.9 (8.1, 11.4) | <0.001ᵃ |
| Encounters per year | 16 (7, 25) | 14 (8, 20) | 0.05 | 14 (6, 28) | 13 (9, 21) | 0.60 |
| Number of encounters | 73 (26, 156) | 124 (73, 205) | <0.001ᵃ | 65 (28, 183) | 140 (84, 210) | <0.001ᵃ |
| Number of unique diagnosis | 68 (36, 113) | 71 (47, 108) | 0.40 | 61 (41, 99) | 73 (47, 103) | 0.20 |

Notes: Continuous variables were reported as median (IQR), and categorical variables were reported as *n* (%). *P*-values were calculated based on two-sided Wilcoxon rank sum test or Pearson's Chi-squared test as appropriate.
ᵃStatistically significant at level 0.05.

bootstrapping (at least 95% of the 1000 iterations), the HLA and AA models identified 28 and 31 risk SNPs, respectively. The top 10 risk SNPs in variable importance selected by each model were shown in Table 3, with a detailed list, including related information, provided in Supplementary Table 2.

By assessing the feature importance of the SNPs chosen by the models, we found that for the HLA GIA group, the top three important predictors were rs429358 (chr19:44908684, nearest gene: *APOE*), rs2075650 (chr19:44892362, nearest gene: *TOMM40*), and rs483082 (chr19: 44912921, nearest gene: *APOC1*), which together accounted for ~25% of the total predictive importance. For the AA GIA group, the most influential predictors were rs2627641 (chr19:45205500, nearest gene: *BLOC1S3*), rs8073976 (chr17:44955857, nearest gene: *C1QL1*), and rs77283277 (chr7: 143386852, nearest gene: *ZYX*).

Eight risk SNPs were identified by both GIA Elastic Net SNPs models, including two AD-associated SNPs (rs429358 and rs2075650) from the top 10 features in both GIA groups, though their relative importance varied slightly. Both models also identified several unique risk SNPs associated with Parkinson's disease (PD), Progressive Supranuclear Palsy (PSP), and stroke as significant predictors of dementia. Notably, the AA GIA model highlighted the significance of a PSP-associated risk SNP, rs8073976, located on chromosome 17, underscoring the distinct genetic underpinnings influencing these different ancestry groups. These findings suggest that while there are common genetic markers associated with dementia across different ancestry groups, there are also unique genetic risk factors that could provide insights into the specific genetic architecture and risk profiles of dementia in diverse populations.

To better understand the biological functions and pathways associated with the identified risk variants, we mapped these risk SNPs to genes using FUMA, which integrates positional, eQTL, and 3D chromatin mapping[31].

Notably, 13 genes were identified by both non-European GIA models (Fig. 2 and Supplementary Table 3). Most shared genes were located near *chr19q13*, which includes the well-established AD risk gene cluster, *APOE-TOMM40-APOC1*[32]. According to the enrichment analysis, these shared genes are predominantly involved in biological pathways associated with lipid metabolism. These pathways encompass processes such as the assembly and organization of protein-lipid complexes, as indicated by the Gene Ontology (GO) terms. Additionally, these genes play an essential role in regulating cholesterol, triglyceride, amyloid proteins, and lipoprotein particles, highlighting the importance of lipid metabolic processes in dementia. There are also shared genes located near *chr3p22* (*SLC25A38* and *RPSA*, PSP risk genes), *chr11q25* (*IGSF9B* and *JAM3*, PD risk genes) and *chr17q21* (*CCDC43*, PSP risk gene).

In addition, we investigated ancestry-specific genes. For instance, genes near the *chr4p16* (e.g., *PCGF3* and *RP11-67M1.1*) were uniquely pinpointed by the HLA GIA model, while genes near the *chr7q34* region (e.g., *ZYX* and *ARHGEF5*) were uniquely identified by the AA GIA model. This differentiation underscores the unique genetic architecture influencing dementia

risk across different ancestry groups and suggests potential pathways for tailored interventions.

In the sensitivity analyses, we performed dementia risk modeling in the EAA GIA sample (N = 673). Similar to other GIA groups, the model incorporating gene-annotated SNPs from AD and other dementia-related disease GWASs performed the best compared to all other models. This model enhanced the AUPRC by 43% (0.34 [95%CI: 0.24, 0.43] vs. 0.24 [95% CI: 0.19, 0.29], Wilcoxon test *p*-value < 0.001) and the AUROC by 12% (0.80 [95%CI: 0.73, 0.86] vs. 0.71 [95%CI: 0.68, 0.74], DeLong test *p*-value = 0.001) compared to the best PRS model. It also outperformed the *APOE-ε4* count model, with increments of 42% in AUPRC (0.34 vs. 0.24 [95%CI: 0.19, 0.30], Wilcoxon test *p*-value < 0.001) and 11% in AUROC (0.80 vs. 0.71 [95%CI: 0.68, 0.73], DeLong test *p*-value = 0.004).

Among the featured 16 risk SNPs, rs429358 (chr19:44908684, nearest gene: *APOE*), rs66626994 (chr19:44924977, nearest gene: *APOC1P1*), and rs6857 (chr19:44888997, nearest gene: *NECTIN2*) were the most significant predictors for the EAA GIA group, collectively accounting for ~45% of the overall predictive importance. After mapping these SNPs to gene, we identified the AD-risk gene cluster, *APOE-TOMM40-APOC1*, as well as the gene region near *chr17q21* (e.g., *FMNL1* and *SPPL2C*) (Supplementary Table 4A-D).

## Validations in the All of Us sample

We conducted a validation study using the All of Us cohort to evaluate the broad applicability of our findings obtained from the UCLA ATLAS sample. A comparable sample was selected from the All of Us Research Hub, employing the same selection scheme to their corresponding GIA groups in the UCLA ATLAS sample. However, due to the limited number of eligible dementia cases (N case = 8) in the All of Us EAA GIA sample, we could only validate our models and findings in the HLA (N_case = 68, N_control = 390) and AA (N_case = 129, N_control = 516) samples. In contrast to the UCLA ATLAS samples, participants in the All of Us cohort had shorter durations of EHR documentation and fewer recorded healthcare visits. The prevalence of dementia was also lower in the All of Us cohort in the HLA GIA group. Within each GIA sample, we found similar distributions of demographics and EHR features between dementia cases and eligible controls (Supplementary Tables 5–6).

We applied the model weights trained from the UCLA ATLAS sample to the All of Us sample, stratified by GIA groups. In comparing three representative models – (1) the *APOE-ε4* model; (2) the best-performing PRS model; and (3) the best-performing Elastic Net SNP model – and accounting for demographic variables (age and sex) and genetic population structure (ancestry-specific PCs), our results mirrored those from the UCLA ATLAS sample. The Elastic Net SNP model, which included gene-annotated SNPs from GWASs of AD and other dementia-related diseases, outperformed all other models in terms of the AUPRC and AUROC in both the HLA and AA GIA samples (Table 4).

**Table 2 | Model performance of *APOE-ε4* count, polygenic risk score, and SNP models in dementia genetic prediction, UCLA ATLAS sample, stratified by genetic inferred ancestry[a]**

| | | AUPRC | AUROC | F1 score | Accuracy | Precision | Recall | Specificity |
|---|---|---|---|---|---|---|---|---|
| Hispanic Latino Americans (*N* = 610) | | | | | | | | |
| APOE | ε4 count | 0.308 | 0.652 | 0.424 | 0.707 | 0.357 | 0.524 | 0.754 |
| AD-PRS models | | | | | | | | |
| AD EUR PRS | P-significant | 0.306 | 0.619 | 0.335 | 0.759 | 0.389 | 0.294 | 0.880 |
| | Gene-annotated | 0.288 | 0.615 | 0.387 | 0.397 | 0.245 | 0.921 | 0.260 |
| AD AFR PRS | P-significant | 0.312 | 0.644 | 0.409 | 0.692 | 0.339 | 0.516 | 0.738 |
| | Gene-annotated | 0.305 | 0.648 | 0.427 | 0.666 | 0.330 | 0.603 | 0.682 |
| AD multi-ancestry PRS | P-significant | 0.298 | 0.626 | 0.389 | 0.444 | 0.252 | 0.857 | 0.337 |
| | Gene-annotated | 0.298 | 0.640 | 0.401 | 0.448 | 0.259 | 0.897 | 0.331 |
| Multi-PRS models | | | | | | | | |
| PRSs using AD GWASs only[b] | P-significant | 0.312 | 0.643 | 0.415 | 0.644 | 0.314 | 0.611 | 0.653 |
| | Gene-annotated | 0.302 | 0.646 | 0.404 | 0.670 | 0.322 | 0.540 | 0.705 |
| PRSs using AD + Neuro GWASs[c] | P-significant | 0.283 | 0.617 | 0.382 | 0.661 | 0.306 | 0.508 | 0.700 |
| | Gene-annotated | 0.309 | 0.643 | 0.411 | 0.662 | 0.321 | 0.571 | 0.686 |
| Elastic Net SNPs models | | | | | | | | |
| SNPs from AD GWASs only | P-significant | 0.321 | 0.662 | 0.408 | 0.530 | 0.276 | 0.786 | 0.463 |
| | Gene-annotated | 0.351 | 0.679 | 0.436 | 0.602 | 0.308 | 0.746 | 0.564 |
| SNPs from AD + Neuro GWASs | P-significant | 0.359 | 0.715 | 0.472 | 0.633 | 0.336 | 0.794 | 0.591 |
| | Gene-annotated | 0.410 | 0.728 | 0.458 | 0.779 | 0.463 | 0.452 | 0.864 |
| Non-linear SNPs models | | | | | | | | |
| SNPs from AD + Neuro GWASs | GBM | 0.304 | 0.634 | 0.381 | 0.707 | 0.337 | 0.437 | 0.777 |
| Gene-annotated SNPs | XGBoost | 0.298 | 0.642 | 0.375 | 0.710 | 0.338 | 0.421 | 0.785 |
| African Americans (*N* = 440) | | | | | | | | |
| APOE | ε4 count | 0.271 | 0.606 | 0.388 | 0.570 | 0.267 | 0.714 | 0.537 |
| AD-PRS models | | | | | | | | |
| AD EUR PRS | P-significant | 0.221 | 0.592 | 0.369 | 0.432 | 0.234 | 0.869 | 0.329 |
| | Gene-annotated | 0.226 | 0.573 | 0.348 | 0.318 | 0.213 | 0.952 | 0.169 |
| AD AFR PRS | P-significant | 0.242 | 0.584 | 0.322 | 0.732 | 0.311 | 0.333 | 0.826 |
| | Gene-annotated | 0.241 | 0.581 | 0.344 | 0.584 | 0.246 | 0.571 | 0.587 |
| AD multi-ancestry PRS | P-significant | 0.234 | 0.592 | 0.360 | 0.386 | 0.225 | 0.905 | 0.264 |
| | Gene-annotated | 0.230 | 0.598 | 0.370 | 0.443 | 0.236 | 0.857 | 0.346 |
| Multi-PRS models | | | | | | | | |
| PRSs using AD GWASs only[b] | P-significant | 0.238 | 0.589 | 0.358 | 0.527 | 0.242 | 0.690 | 0.489 |
| | Gene-annotated | 0.233 | 0.590 | 0.357 | 0.484 | 0.234 | 0.750 | 0.421 |
| PRSs using AD + Neuro GWASs[c] | P-significant | 0.187 | 0.516 | 0.311 | 0.195 | 0.186 | 0.952 | 0.017 |
| | Gene-annotated | 0.217 | 0.538 | 0.087 | 0.809 | 0.500 | 0.048 | 0.989 |
| Elastic Net SNPs models | | | | | | | | |
| SNPs from AD GWASs only | P-significant | 0.356 | 0.669 | 0.356 | 0.802 | 0.471 | 0.286 | 0.924 |
| | Gene-annotated | 0.421 | 0.678 | 0.342 | 0.834 | 0.704 | 0.226 | 0.978 |
| SNPs from AD + Neuro GWASs | P-significant | 0.391 | 0.704 | 0.342 | 0.825 | 0.606 | 0.238 | 0.963 |
| | Gene-annotated | 0.446 | 0.710 | 0.365 | 0.834 | 0.677 | 0.250 | 0.972 |
| Non-linear SNPs models | | | | | | | | |
| SNPs from AD + Neuro GWASs | GBM | 0.225 | 0.479 | 0.314 | 0.186 | 0.187 | 0.976 | 0.000 |
| Gene-annotated SNPs | XGBoost | 0.220 | 0.506 | 0.139 | 0.802 | 0.412 | 0.083 | 0.972 |

Abbreviations: *AD* Alzheimer's Disease, *APOE* apolipoprotein E, *AUROC* Area Under the ROC Curve, *AUPRC* Area Under the Precision-Recall Curve, *EUR* European, *GBM* Gradient Boosting Machine, *GWAS* Genome-Wide Association Study, *PRS* Polygenic Risk Score, *SNP* Single-Nucleotide Polymorphism.
Notes:
[a]All models (if not other specified) have regressed out age, sex, and ancestry-specific principal components. Thresholds were determined by maximizing absolute Matthews correlation coefficient.
[b]All AD PRSs built with EUR, AFR, and multi-ancestry GWASs using P-significant/gene-annotated SNPs were included in the model at the same time.
[c]All AD PRSs built with EUR, AFR, and multi-ancestry GWASs, and neurodegenerative disease PRS (Parkinson's disease, progressive supranuclear palsy, Lewy body dementia, and stroke) using P-significant/gene-annotated SNPs were included in the model at the same time.

**Table 3 | Top 10 featured risk SNPs from the best-performing Elastic Net SNP model, UCLA ATLAS sample, stratified by genetic ancestry**

| rsID | CHR | POS | Variable Importance (95% CI) | Nearest Gene | AD EUR | AD AFR | AD multi | LBD | PD | PSP | Stroke |
|------|-----|-----|------------------------------|--------------|--------|--------|----------|-----|-----|-----|--------|
| **Hispanic Latino American ancestry (HLA)** | | | | | | | | | | | |
| **rs429358** | 19 | 44908684 | 0.089 (0.02, 0.11) | *APOE* | | x | | | | | |
| **rs2075650** | 19 | 44892362 | 0.089 (0.02, 0.11) | *TOMM40* | | x | x | x | | | |
| rs483082 | 19 | 44912921 | 0.07 (0.017, 0.086) | *APOC1* | | x | x | | | | |
| rs157581 | 19 | 44892457 | 0.065 (0.014, 0.079) | *TOMM40* | | x | | x | | | |
| rs412776 | 19 | 44876259 | 0.057 (0.016, 0.066) | *NECTIN2* | x | | x | | | | |
| rs62120578 | 19 | 44713297 | 0.053 (0.022, 0.061) | *CTB-171A8.1* | x | | | | | | |
| rs4803765 | 19 | 44855191 | 0.048 (0.015, 0.056) | *NECTIN2* | x | | | | | | |
| rs80100206 | 4 | 705856 | 0.042 (0.02, 0.046) | *PCGF3* | | | | | x | | |
| rs6857 | 19 | 44888997 | 0.038 (0.011, 0.042) | *NECTIN2* | | x | | | | | |
| rs2276412 | 11 | 121590137 | 0.037 (0.018, 0.039) | *SORL1* | x | | | | | | |
| **African American ancestry (AA)** | | | | | | | | | | | |
| rs2627641 | 19 | 45205500 | 0.096 (0.077, 0.099) | *BLOC1S3* | x | | | | | | |
| rs8073976 | 17 | 44955857 | 0.079 (0.065, 0.082) | *C1QL1* | | | | | | x | |
| rs77283277 | 7 | 143386852 | 0.076 (0.063, 0.079) | *ZYX* | x | | | | | | |
| **rs429358** | 19 | 44908684 | 0.071 (0.059, 0.074) | *APOE* | | x | | | | | |
| **rs2075650** | 19 | 44892362 | 0.068 (0.056, 0.071) | *TOMM40* | | x | x | x | | | |
| rs73936967 | 19 | 44890485 | 0.064 (0.053, 0.066) | *TOMM40* | | x | | | | | |
| rs13032148 | 2 | 127107524 | 0.063 (0.053, 0.065) | *BIN1* | x | | x | | | | |
| rs71352239 | 19 | 44926286 | 0.056 (0.047, 0.057) | *APOC1P1* | x | | x | x | | | |
| rs435380 | 19 | 44903861 | 0.041 (0.037, 0.042) | *TOMM40* | | x | x | | | | |
| rs11223641 | 11 | 133950127 | 0.041 (0.037, 0.041) | *IGSF9B* | | | | | x | | |

*AD* Alzheimer's Disease, *AFR* African American, *CI* confidence interval, *EUR* European, *LBD* Lewy body dementia, *PD* Parkinson's disease, *PRS* Polygenic Risk Score, *PSP* progressive supranuclear palsy, *SNP* Single-Nucleotide Polymorphism.
Note: SNPs marked in bold are overlapped SNPs identified by both samples.

**Fig. 2 | Shared and ancestry-specific risk genes identified by the best-performing Elastic Net SNP models, UCLA ATLAS sample.**

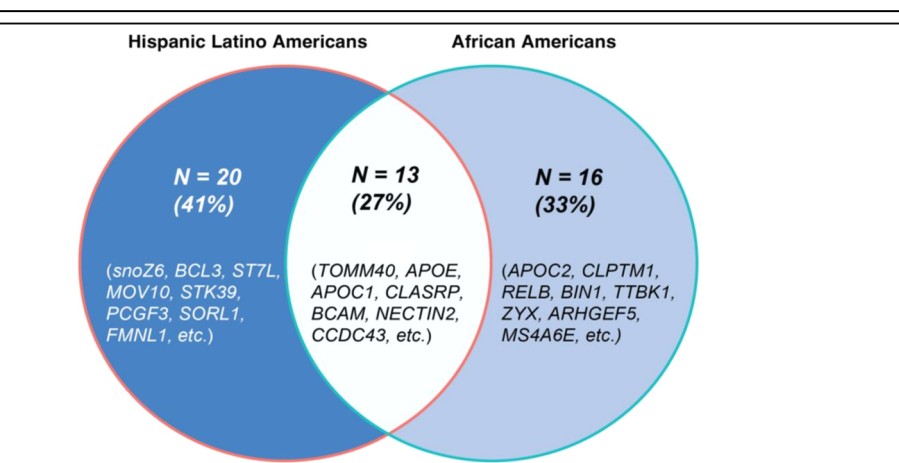

| Gene sets | |
|-----------|---|
| **Shared** | *APOC1, APOE, BCAM, CCDC43, CLASRP, CTB-171A8.1, IGSF9B, JAM3, PVRL2, RP11-713P17.5, RPSA, SLC25A38, TOMM40* |
| **HLA specific** | *ARHGAP27, ARHGAP27, BCL3, CTB-39G8.2, CTB-39G8.3, FMNL1, KANSL1-AS1, MAP3K14, MOV10, PCGF3, PLEKHM1, RP11-67M1.1, RP11-67M1.1, RP11-777N19.1, SMARCA4, snoZ6, SORL1, SPATA32, SPPL2C, ST7L, STK39, WNT2B* |
| **AA specific** | *AC006126.4, ADAM11, APOC2, ARHGEF5, BIN1, C1QL1, CBLC, CLPTM1, CTB-129P6.4, CTD-2534I21.9, DBF4B, GFAP, MS4A6E, RELB, TTBK1, ZYX* |

**Table 4 | Overall model performance of *APOE-ε4* count, polygenic risk score, and Elastic Net SNP models in dementia genetic prediction in validation of All of Us sample, stratified by genetic inferred ancestry**

|  |  | HLA (*N* = 458) | | AA (*N* = 645) | |
|---|---|---|---|---|---|
|  |  | Cases | Controls | Cases | Controls |
|  | *N* | 68 | 390 | 129 | 516 |
| **Model** |  | **AUPRC** | **AUROC** | **AUPRC** | **AUROC** |
| *APOE* | ε4 count | 0.206 | 0.601 | 0.211 | 0.588 |
| Best single AD PRS | AFR P-significant | 0.231 | 0.593 | 0.253 | 0.524 |
| Best SNPs | Gene-annotated Neuro SNPs | 0.240 | 0.617 | 0.285 | 0.639 |

*AA* African Americans, *AD* Alzheimer's Disease, *AFR* African American, *APOE* apolipoprotein E, *AUROC* Area Under the ROC Curve, *AUPRC* Area Under the Precision-Recall Curve, *HLA* Hispanic Latino Americans, *PRS* Polygenic Risk Score, *SNP* Single-Nucleotide Polymorphism.

In particular, the Elastic Net SNP model demonstrated significant improvements over the other two models. In the HLA GIA sample, it outperformed both the *APOE-ε4* and the best AD PRS model (AD AFR *PRS.psig*) by 17% and 4% in AUPRC (both Wilcoxon test *p*-value < 0.001), and by 2.7% and 4% in AUROC (DeLong test *p*-value = 0.56 and 0.25), respectively. Similarly, in the AA GIA sample, the Elastic Net SNP model showed a 35% and 13% enhancement in AUPRC (both Wilcoxon test *p*-value < 0.001), and 9% and 22% in AUROC (both DeLong test *p*-value < 0.001) over the *APOE-ε4* and best AD PRS model, respectively.

## Discussion

Traditional genetic risk models have faced limitations in effectively capturing causal disease risk variants and accurately assessing genetic risks across diverse populations. To address these challenges, our present study introduces a novel approach to predicting dementia risks by leveraging functional mapping of genetic data in conjunction with machine learning methods in the real-world EHR setting. Our proposed method shows remarkable improvements in prediction performance compared to well-known approaches like *APOE* gene and PRS models. We successfully identified shared and ancestry-specific risk genes and biological pathways contributing to dementia risks for each non-European GIA group. Finally, we bolstered the reliability and generalizability of our findings by validating our models using a comparable EHR sample from the All of Us cohort.

Our study highlights the significance of prioritizing biologically meaningful SNPs in genetic prediction. GWASs often identify genomic regions with multiple correlated SNPs, which may encompass several closely located genes. However, not all of these genes are relevant to the disease[33]. Functional annotation of genetic variants enabled us to target potential causal SNPs by considering various factors, such as regional linkage disequilibrium (LD) patterns, functional consequences of variants, their impact on gene expression, and their involvement in chromatin interaction sites[31]. In our models developed on UCLA ATLAS samples, we achieved significant improvements in model performance by prioritizing biologically meaningful SNPs, ranging from 31–84% in AUPRC and 11–17% in AUROC across different GIA groups, compared to the *APOE-ε4* count and the best-performing PRS models. These results underscore the critical role of considering functional and biological information in enhancing the performance of genetic prediction models, especially in diverse populations.

It is worth highlighting that no discernible performance differences were observed between PRSs constructed using genome-wide-significant and gene-annotated SNPs. This can be attributed to the strong LD between genome-wide-significant and gene-annotated SNPs within the same genomic region. As a result, these SNPs tend to have similar effect estimates in the GWASs. Thus, it is expected that the PRSs built with these two sets of SNPs would exhibit a high correlation (Supplementary Table 7), which

further supports the notion that the choice of genome-wide-significant or gene-annotated SNPs does not significantly impact the predictive performance of the PRSs in our study.

Moreover, our study emphasizes the significance of incorporating risk factors from multiple dementia-related diseases when developing predictive models for complex conditions like dementia. Both ancestry-specific Elastic Net SNP models highlighted several PD and PSP risk variants as significant predictors of dementia. This finding aligns with the well-known complexity of dementia as a multifactorial disorder that shares common features with these related conditions[34]. However, it is worth noting that including PRSs of those diseases did not significantly improve the overall performance (Table 2). This result is consistent with research conducted by Clark et al.[35], in which they demonstrated that a combined genetic score, which incorporated risk variants for AD and 24 other traits, had an equivalent predictive power as the AD PRS on its own.

Our proposed Elastic Net SNPs models identified several shared risk factors across different ancestries. Notably, a substantial proportion of the identified shared genes were found near the *chr19q13* region, which is well-known for the AD risk gene cluster comprising *APOE-TOMM40-APOC1*. These findings align with previous research[6,36,37], further supporting the significance of this genomic region in contributing to the genetic risks associated with dementia.

At the same time, we have discovered compelling evidence supporting our hypothesis that risk SNPs associated with dementia, along with their corresponding weights, exhibit significant variations across diverse populations. Notably, our analysis of PRS models revealed that the performance of PRS built with the European population GWAS was worse when predicting a non-European GIA group. This is consistent with other previous studies. Using PRSs for 245 curated traits from the UK Biobank data, Privé et al.[38] revealed notable disparities in the phenotypic variance explained by PRSs across different populations. Specifically, compared to individuals of Northwestern European ancestry, the PRS-driven phenotypic variance is only 64.7% in South Asians, 48.6% in East Asians, and 18% in West Africans. Similarly, using a population from the Health and Retirement Study, Marden et al. demonstrated that the estimated effect of the AD PRS was notably smaller for non-Hispanic black compared to non-Hispanic white in both dementia probability score and memory score[39]. On the other hand, we also observed that the *APOE-ε4* count model performed better than most PRS models in HLA and AA GIA samples. These finding further reinforces the limitations of standard PRS when applied to non-European populations, in which attempting to transfer GWAS effect size from one GIA to another GIA, or when using matched genetic ancestry GWAS with smaller sample size, as demonstrated in several AD and other phenotype studies[40–43].

In addition, we observed notable differences in the feature importance of various SNPs within the best-performing Elastic Net models across distinct GIA groups. Consequently, this led us to identify ancestry-specific genes and distinct biological pathways implicated in the genetic predisposition to dementia in diverse ancestral samples. These findings highlight the uniqueness of genetic risk factors and functional pathways in diverse population groups.

Finally, we validated our models using samples from separate EHR linked with genetic data (All of Us). Our proposed Elastic Net SNP model consistently outperformed the *APOE-ε4* and the best PRS models. While the Elastic Net SNP model demonstrated improved performance in both HLA and AA populations, we observed a decrease in the general performance and significance (AUPRC and AUROC) in the All of Us sample compared to the UCLA ATLAS sample. One potential explanation for this discrepancy is the distinct population structure within each sample, as revealed by comparing patient characteristics (Supplementary Table 6). These findings underscore the influence of population-specific factors within GIA groups on the generalizability of genetic risk models, highlighting the critical need to account for population diversity in predictive models for complex diseases.

Our study boasts several notable strengths that contribute to its significance and impact. Firstly, we conducted our research with EHRs that mirror the practicalities of real-world community settings. Most current

studies used longitudinal cohorts, which performed extensive testing and consensus criteria[44] applied by clinicians with expertize in dementias to diagnosis dementia. However, in real-world clinical care, the expertize in dementia may vary, and the criteria used for diagnosis may not always align with the stringent standards of research cohorts. Diagnoses documented in the EHRs capture these real-world data and, by routinely capturing patient data over extended periods, form an expansive longitudinal cohort ideal for real-world research. Compared to traditional cohorts, EHR cohorts offer additional benefits, such as vast sample sizes, diverse phenotypes, and a more inclusive representation of often underrepresented groups, like minority groups and older adults[45]. Secondly, machine learning techniques applied in our study allowed us to infer crucial dementia risk factors for underrepresented populations, such as HLA and AA, with GWAS summary statistics from extensively studied populations like Europeans. This approach enabled a deeper understanding of the genetic landscape of dementia in underrepresented populations, particularly valuable given the current limitations in large-sample-size GWASs specific to these groups. Thirdly, we fortified the robustness and generalizability of our findings through the validation of our model on an independent dataset from the All of Us cohort. Furthermore, our innovative approach, which incorporated biologically relevant genetic markers and functional annotations, significantly enhanced the accuracy of disease prediction. This approach can be readily adapted to predict other complex diseases, extending the scope of its applications and enriching our understanding of diverse human populations' genetic traits.

However, we acknowledge certain limitations. Firstly, we observed variations in the composition of dementia subtypes among different GIA groups' case samples. Consequently, the distinct genes and biological pathways identified by different ancestry models should be interpreted with this consideration. Secondly, although our study identified potential risk SNPs and genes associated with dementia, additional experimentation is necessary to understand the precise mechanisms underlying the association of these factors. Thirdly, the limited number of dementia cases in our non-European GIA samples, after applying inclusion criteria, constrains the generalizability of our findings. Future studies should aim to replicate these findings in larger samples for each GIA to enhance their robustness. Finally, although detailed clinical guidelines for disease diagnoses exist[46,47], clinical providers may adapt these criteria to fit specific research focuses or populations. This adaptation can lead to variations in diagnostic criteria across different studies or clinical practices. Consequently, the precision of dementia diagnoses based on ICD-10 codes may vary compared to a gold standard of research criteria or autopsy findings.

In light of these limitations, further research with more extensive and diverse datasets, encompassing a broader range of dementia subtypes and GIA groups is imperative to strengthen the validity and applicability of our study's outcomes. Such efforts will contribute to a more comprehensive understanding of the genetic complexities underlying dementia across diverse populations.

In conclusion, our study introduces a novel and robust approach to assessing individual genetic risks for dementia across diverse populations in a real-world setting. Our study demonstrates the importance of considering functional and biological information and population diversity when developing predictive models for complex diseases like dementia. The findings from our research provide valuable insights into the intricate genetic factors underlying dementia. Moreover, this work opens up promising avenues for developing more accurate and efficient predictive models for complex genetic traits in diverse human populations. Such advancements can potentially be paired with the development of targeted treatments tailored to the specific genetic profiles of individuals affected by dementia and related conditions.

## Methods
### Data source
Our discovery cohort for model development was derived from the biobank-linked EHR of the UCLA Health System[29]. The UCLA ATLAS

Community Health Initiative collects biosamples during routine lab work at UCLA Health labs from a diverse population, which undergoes genotyping using a customized Illumina Global Screening Array[48]. Participants watch a short video explaining the initiative's goals and record their consent decision. Detailed information regarding biobanking and consenting procedures is available in our previous publications[49,50]. After the genotype quality control, there were 54,935 individuals with both genotype and UCLA EHR data. All ethical regulations relevant to human research participants were followed. As the genetic data and EHRs were de-identified, the study was exempt from human subject research regulations (UCLA IRB# 21-000435).

We validated our models and findings using data from the All of Us Research Hub, one of the most diverse biomedical data resources in the United States. The All of Us Research Program serves as a centralized data repository, offering secure access to de-identified data from program participants[51]. For validation, we utilized data release version 7, encompassing 409,420 individuals, of which 245,400 have undergone whole genome sequencing.

### Patient genetic data preprocessing
Quality control was conducted using PLINK v1.9[52], adhering to established guidelines[29]. Samples with a missingness rate exceeding 5% were removed. Low-quality SNPs with >5% missingness, monomorphic SNPs, and strand-ambiguous SNPs were excluded. Post-quality control, genotype imputation was performed via the Michigan Imputation Server[53] to enhance the coverage of genetic variants and facilitate comparison across diverse genotyping platforms. SNPs with imputation $r^2 < 0.90$ or minor allele frequency <1 % were pruned. After these measures, 21,220,668 genotyped SNPs were retained across the 54,935 individuals. Finally, we restricted our analyses to SNPs that overlapped between UCLA ATLAS and All of Us, resulting in a total of 8,705,988 SNPs, ensuring consistency in genetic variables across datasets.

Genetic ancestry refers to the geographic origins of an individual's genome, tracing back to their most recent biological ancestors[54]. GIA employs genetic data, a reference population, and inferential methodologies to categorize individuals within groups likely sharing common geographical ancestors[55]. In our UCLA ATLAS sample, we used the reference panel from the 1000 Genomes Project[56] and principal component analysis[57] to infer genetic ancestry. GIA groups included European American (EA), African American (AA), Hispanic Latino American (HLA), East Asian American (EAA), and South Asian American (SAA). For instance, individuals in the United States with recent biological ancestors inferred to be of Amerindian ancestry were designated as "HLA GIA"[58]. In addition, we calculated ancestry-specific principal components within each GIA group using principal component analysis.

### Genetic predictors
The initial step in our study involved identifying potential risk SNPs as candidate predictors for dementia using GWASs. A summary of the GWASs used and the steps to select candidate SNPs is provided in Supplementary Table 8 and Supplementary Fig. 2.

We selected GWASs for AD[5,36,59], Parkinson's disease (PD)[60], Progressive Supranuclear Palsy (PSP)[61], Lewy Body Dementia (LBD)[62], and stroke[63] phenotypes. For AD, we included three GWASs conducted on diverse populations, including European[5], African American[36], and multi-ancestries[59]. Summary statistics from these GWAS are publicly available, with detailed recruitment procedures and diagnostic criteria available in the original publications.

A significant proportion of GWAS hits are located in non-coding or intergenic regions[64]. Due to the correlated nature of genetic variants in LD, distinguishing causal from non-causal variants based solely on association P-values from GWASs is challenging[31]. Identifying the most likely causal variants involves understanding the regional LD patterns and assessing the functional consequences of correlated SNPs, such as those affecting protein-coding, regulatory, and structural sequences[65]. Several functionally validated variants have been clinically relevant to diseases pathogenesis, confirmed

**Fig. 3 |** Sample selection steps and modeling steps description.

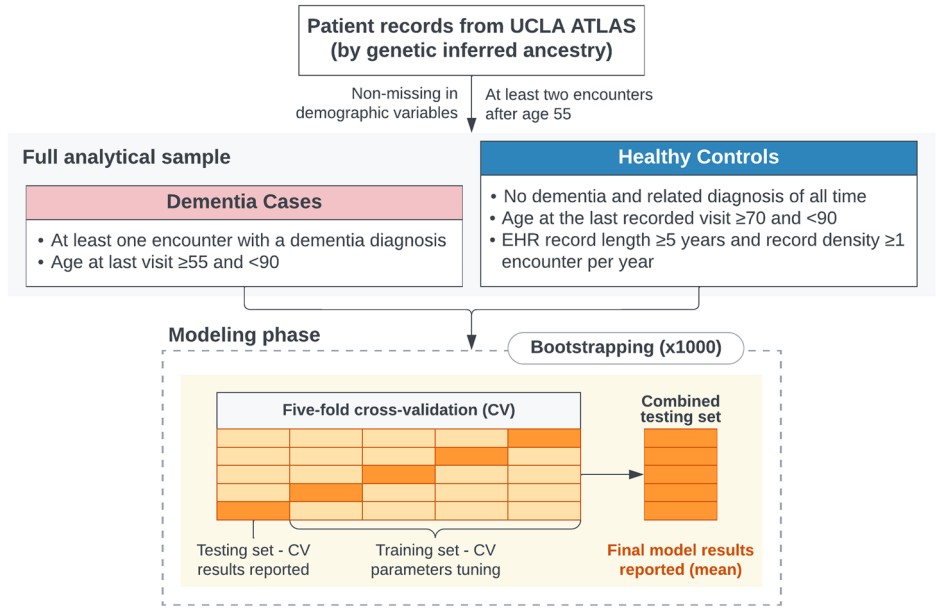

through experimental validation[66]. To address this, we utilized the FUMA, a tool that leverages information from biological data repositories to annotate and prioritize SNPs[31].

For each GWAS summary statistic, we first identified genomic risk loci using a *P*-value threshold (<5e-8) and a pre-calculated LD structure ($r^2 < 0.2$) based on the relevant reference population from the 1000 Genomes Project[56]. Subsequently, we identified two distinct sets of SNPs:

1. **Independent genome-wide-significant SNPs**: We selected the SNP with the most significant GWAS *P*-value within each genomic risk locus. This process was iterated until all SNPs were assigned to a risk locus cluster or considered independent.

2. **Independent gene-annotated SNPs**: We prioritized SNPs based on their functional consequences on genes. Using FUMA, the mapping from SNPs to genes was achieved by performing ANNOVAR[67] using Ensembl genes (build 85). SNPs were mapped to genes through positional mapping, eQTL associations, and 3D chromatin interactions. The Combined Annotation-Dependent Depletion (CADD) score[68] was used to select potential causal SNPs, with the SNP possessing the highest CADD score within each genomic risk locus being chosen, indicating a higher probability of the variant being deleterious.

The identified independent genome-wide-significant SNPs and independent gene-annotated SNPs were subsequently used in constructing the disease PRSs and as candidate features in dementia prediction models. To ensure robustness in a sensitivity analysis, we also adopted a stringent $r^2$ cut-off (< 0.1) to define independent genome-wide-significant SNPs, ensuring the selected SNPs were independent.

We computed the disease-specific PRS as the sum of an individual's risk allele dosages, each weighted by its corresponding risk allele effect size from the GWAS summary statistics, as shown in the PRS equation $PRS_i = \sum_j^M \hat{\beta}_j \times dosage_{ij}$. All PRSs were standardized to a mean of 0 and a standard deviation of 1. The standardization process used the 1000 Genome European genetic ancestry as the reference population, ensuring the scores' range and values are comparable across different GWASs. For each phenotype, we employed two distinct sets of SNPs identified by FUMA, namely the independent genome-wide-significant SNPs and independent gene-annotated SNPs, to calculate two respective PRSs: *PRS.psig* and *PRS.map*.

The *APOE* gene has two variants, rs7412 and rs429358, which determine the three common isoforms of the apoE protein: E2, E3, and E4, encoded by the ε2, ε3, and ε4 alleles[37]. Previous research has demonstrated

that carriers of *APOE-ε4* are at a higher risk of developing AD, exhibiting a dose-dependent effect[69]. Therefore, to quantify the *APOE* genotype in our study, we created a numerical variable, "*APOE-ε4count*", representing the number of ε4 alleles (0, 1, or 2) carried by each individual.

### Dementia definition and demographic features
The primary outcome of interest was dementia, defined using the ICD-10 codes (Supplementary Table 9). Demographic variables considered included self-reported age and sex. The age of each participant, measured in years, was calculated based on their birth date and encounters dates. For individuals diagnosed with dementia, we determined the age at dementia onset.

### Analytical sample selection
To focus on patients with longitudinal records, our analyses included patients with complete demographic data (age and sex) who had at least two medical encounters after age 55. We restricted the age at the last recorded encounter to <90, as patients in the UCLA EHR dataset are censored at this age.

Eligible dementia cases were identified as patients with at least one encounter with a recorded dementia diagnosis, provided the initial onset occurred after age 55. Eligible controls were required to meet the following criteria: (1) no recorded dementia or related diagnoses, as determined by predefined exclusion phenotypes[70]; (2) age at the last recorded visit ≥ 70, to exclude younger patients who may not have manifested signs of dementia yet; and (3) a minimum of 5 years of records with an average of at least one encounter per year, minimizing potential bias from mis-diagnosis (Fig. 3).

### Prediction of dementia risk with machine learning models
In our discovery study, we developed machine learning models to predict the binary dementia phenotype in the UCLA ATLAS sample, stratified by GIA groups.

To distinctly assess genetic influences, our analysis began by mitigating the impact of demographic factors, including age, sex, and ancestry-specific PCs. We first employed a logistic regression model that utilized only these variables to predict dementia status. Subsequently, we derived the predicted values for each patient through this model. Applying an appropriate inverse link function (e.g., logit), we then subtracted these predicted values from the ultimate outcome (dementia status), generating an "offset" value. These offset values encapsulated the dementia status after regressing out the effects of demographic variables and genetic population structure.

Next, we trained genetic risk models to predict dementia status with offset corrections applied in the linearized space, expressed as: $\hat{y}_i = g^{-1}(\beta_0 + \beta_1 x_{i1} + \cdots + \beta_p x_{ip} + offset_i)$, where $\hat{y}_i$ represents the predicted dementia status, and $g^{-1}(\cdot)$ is the inverse of the link function[71]. We compared four different sets of predictors: (1) *APOE* status, (2) AD PRS, (3) multiple PRSs, and (4) smaller SNP sets with Elastic Net regularization. For the multiple PRS models, we crafted models utilizing diverse AD PRSs of varying ancestries or PRSs derived from other GWASs focused on neurodegenerative diseases. The (4) model involved the application of Elastic Net regularization, which combines the benefits of both Lasso (L1) and Ridge (L2) regression methods to enhance model stability and variance handling. This technique aids in variable selection by reducing the coefficients of less relevant variables to zero, simplifying the model, and improving its ability to manage multicollinearity[30]. The hyperparameter α, which balances L1 and L2 regularization, was optimized using a grid search to maximize the penalized likelihood within each training set.

As part of our sensitivity analysis, we assessed the performance of various non-linear models incorporating different regularization techniques, including GBM[72] and XGBoost[73]. Hyperparameter optimization was also performed using a grid search approach for each model within each training set.

We employed a 5-fold cross-validation methodology across all models to evaluate performance, with final results reported on the combined holdout testing sets (Fig. 3). To enhance the robustness of our findings, we utilized bootstrapping[74] to determine feature importance, determine confidence intervals (CIs), and establish statistical significance. Specifically, we repeated the modeling process 1000 times using random sampling with replacement of all subjects (cases and controls) within the analytical sample set of each GIA group.

The primary assessment criterion was the AUPRC, chosen for its suitability in scenarios involving imbalanced datasets where the number of cases is significantly outnumbered by controls[75]. Additionally, the AUROC was reported as a comprehensive metric for model evaluation. To determine the optimal threshold, we selected the point that maximized the MCC[45]. Subsequent performance metrics, such as the F1 score, accuracy, precision, recall, and specificity, were computed based on this threshold.

To compare models, we calculated DeLong test *p*-values[76], which are specifically tailored for comparing two AUROC values derived from identical observations. Given the lack of an equivalent test for AUPRC comparisons, we employed the paired Wilcoxon signed-rank test[77] to compare AUPRC using the bootstrapping results.

## Validations in the All of Us sample
We conducted a validation study using the All of Us cohort to assess the generalizability of our findings derived from the UCLA ATLAS sample. A comparable sample was selected, adhering to the same criteria and sampling scheme for the GIA groups as in the UCLA ATLAS sample. We employed the same methodologies to define dementia cases and controls, extracting the same genetic risk loci from the All of Us Whole Genome Sequencing data for PRS construction or those identified through Elastic Net models in the UCLA ATLAS sample. Consistent methodologies were used to regress out demographic variables and genetic population structure (i.e., PCs) to derive offset corrections, ensuring statistical models accurately reflect intrinsic genetic associations without confounding from demographic or population genetic structure.

In the All of Us sample, we compared three models: (1) the *APOE-ε4* model; (2) the best-performing PRS model; and (3) the best-performing Elastic Net SNP model. The same evaluation metrics were utilized for model comparisons.

## Gene mapping and gene set analysis
We further examined the features selected from the Elastic Net SNP models. During bootstrapping, each iteration potentially identified a subset of SNPs as important features contributing to the dementia prediction. SNPs consistently identified in at least 95% of the 1000 bootstrap iterations were retained. To facilitate biological interpretations, we employed FUMA's positional, eQTL, and chromatin interaction mapping to associate dementia risk SNPs from the top-performing Elastic Net SNP models with specific genes[31]. These mapped genes were tested against gene sets procured from MsigDB, including positional gene sets and GO gene sets, to assess the enrichment of biological functions through hypergeometric tests., The Benjamin-Hochberg adjustment was applied to correct for multiple testing[78].

## Statistics and reproducibility
The study included diverse genetic ancestry groups: Hispanic Latino American (610 patients, 126 cases), African American (440 patients, 84 cases), and East Asian American (673 patients, 75 cases). Sample sizes were chosen based on availability and representativeness from UCLA Health records and the All of Us cohort.

Each sample was treated as an independent replicate. Analyses were conducted with appropriate statistical methods to ensure validity and reproducibility. The robustness of the findings was further confirmed through cross-validation techniques and comparison with established models (*APOE* and PRSs).

To ensure reproducibility, we adhered to rigorous data handling and processing standards, with detailed documentation of data sources, processing steps, and analysis pipelines. All codes and scripts used in the analysis are available online and upon request for verification and replication purposes.

## Reporting summary
Further information on research design is available in the Nature Portfolio Reporting Summary linked to this article.

## Data availability
The Genome-Wide Association Study summary statistics data analyzed in this study are publicly available. Individual electronic health record data are not publicly available due to patient confidentiality and security concerns. Collaboration with the study authors who have been approved by UCLA Health for Institutional Review Board-qualified studies are possible and encouraged. Individual data from All of Us are publicly available for qualified researchers per the National Institutes of Health.

## Code availability
Codes are publicly available on GitHub: https://github.com/TSChang-Lab/Dementia-prediction (https://doi.org/10.5281/zenodo.12754446)[79]. Requests for additional information can be directed to the Lead Contact: Timothy S. Chang (timothychang@mednet.ucla.edu).

## List of abbreviations

| Abbr | Description |
| --- | --- |
| AA | African American |
| AD | Alzheimer's disease |
| APOE | Apolipoprotein E |
| AUPRC | area under the precision-recall curve |
| AUROC | area under the receiver operating characteristic |
| CADD | Combined Annotation-Dependent Depletion |
| CI | confidence intervals |
| EA | European American |
| EAA | East Asian American |
| EHR | Electronic Health Record |
| FTD | Frontotemporal dementia |
| FUMA | Functional Mapping and Annotation of Genome-Wide Association Studies |
| GIA | Genetic Inferred Ancestry |
| GO | Gene Ontology |
| GWAS | Genome-Wide Association Studies |

| HLA | Hispanic Latino American |
|---|---|
| LBD | Lewy body dementia |
| LD | Linkage disequilibrium |
| MCC | Matthews Correlation Coefficient |
| PC | principal components |
| PD | Parkinson's disease |
| PRS | Polygenic risk score |
| SAA | South Asian American |
| SNP | Single-Nucleotide Polymorphism |

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

## Acknowledgements

M.F., L.V.B., S.S.W., and T.S.C. was supported by the National Institutes of Health (NIH) National Institute of Aging (NIA) grant K08AG065519-01A1, UH2AG083254, the Hillblom Foundation, and the Fineberg Foundation. T.S.C. was supported by U54NS123746 and the California Department of Public Health. K.V. was supported by NIH grants R01 NS033310, R01 AG058820, R01 AG075955, and R56 AG074473, and UH2 AG083254. B.P. was supported by NIH grants R01HG009120, R01MH115676, and R01HG006399. We gratefully acknowledge the resources provided by the Institute for Precision Health (IPH) and participating UCLA ATLAS Community Health Initiative patients. The UCLA ATLAS Community Health Initiative in collaboration with UCLA ATLAS Precision Health Biobank, is a program of IPH, which directs and supports the biobanking and genotyping of biospecimen samples from participating UCLA patients in collaboration with the David Geffen School of Medicine, UCLA CTSI and UCLA Health. We would also like to acknowledge all participants and researchers at the All of Us program. The All of Us Research Program is supported by the National

Institutes of Health, Office of the Director: Regional Medical Centers: 1 OT2 OD026549; 1 OT2 OD026554; 1 OT2 OD026557; 1 OT2 OD026556; 1 OT2 OD026550; 1 OT2 OD 026552; 1 OT2 OD026553; 1 OT2 OD026548; 1 OT2 OD026551; 1 OT2 OD026555; IAA #: AOD 16037; Federally Qualified Health Centers: HHSN 263201600085U; Data and Research Center: 5 U2C OD023196; Biobank: 1 U24 OD023121; The Participant Center: U24 OD023176; Participant Technology Systems Center: 1 U24 OD023163; Communications and Engagement: 3 OT2 OD023205; 3 OT2 OD023206; and Community Partners: 1 OT2 OD025277; 3 OT2 OD025315; 1 OT2 OD025337; 1 OT2 OD025276.

## Author contributions

M.F., B.P., K.V. and T.S.C. contributed to conception and design of the study. M.F., L.V.B., and S.S.W. performed the statistical analysis. M.F. wrote the first draft of the manuscript. All authors contributed to manuscript revision, read, and approved the submitted version.

## Competing interests

The authors declare no competing interests.
