## [Transparent Peer Review file · Communications Biology]

Improving genetic risk modeling of dementia from real-world data in underrepresented populations

Corresponding Author: Dr Timothy Chang

Figures originally included in the author's rebuttal have been redacted from this file.

Version 0:

Reviewer comments:

Reviewer #1

(Remarks to the Author)

Comments for the Authors

Fu and his/her colleagues explored how genetic risk modelling provide additional benefit to dementia. By integrating SNPs identified from pre-GWAS analysis, authors leveraged Elastic Net to develop prediction models and compare with APOE and PRS to demonstrate the additive values across genetic ancestry groups. The study design and objective are straight forward and easy to follow. The study cohort of UCLA EHR data (discover cohort) and All of US (external validation cohort) are relative large-scaled enough to support the findings. Overall, I think this is an interesting and necessary study; however, and I have several suggestions pertaining to the methodology that hope to improve the study.

Major comments:

1. Why setting case-control ratio in this study as 1:3? Your results of Precision, recall, specificity and accuracy that derived from 2-by-2 contingency matrix were highly depend on the ratio of cases versus controls. In such 1:1, 1:2, ..., 1:5 would largely impact your results. Since your data is a community-based cohort, and it has much more controls, why not put all of them in your study?
2. Elastic Net is actually a mixture of lasso and ridge, two regularized methods on top of regressions, please specific how did you determine the hyperparameter alpha of the proportion between L1 and L2 regularizations. How many SNPs remained in the final model after regularization?
3. Could you please provide more explanation about why you chose the Elastic Net, rather than other models to identify the variables that are associated with different dementia statutes, it would be nice to provide more comparable results of other models.
4. The authors employed a 5-fold cross-validation methodology, how it performed? Did you tune your parameters using the CV scheme? If so, I believe your model might be overfitting as your testing data was used for selection of SNPs. You can only optimize your model using the training data, while the testing need to held out and merely used for evaluation. Please be careful and clearly state this part.
5. The cross-validation and 1000 permutation confused me. CV means you train and evaluate your model separately and iteratively, how this combined with permutation? Did you combined predicted probabilities from each testing fold of data and then evaluated through permutation? I believe a supplementary figure of flowchart could better to illustrate your training and evaluation procedures.
6. The permutation seems exact the bootstrap method, and your p-value was based on a hypothesis that the metrics used are simple sums of IID random variables. This is not a trivial problem as the hypothesis not stand for such situation, and comparing them additionally requires taking into account the correlation between metrics calculated on the same sample. Please consider DeLong statistics for AUROC.
7. The authors defined dementia, AD, VD, etc based on ICD-10, I am curious about what the diagnostic criteria for these disease? Are they following the same criteria of UCLA and All of US cohort?

Minor comments:

1. Please add the number of participants, target cases to your abstract.
2. Also, please add the range of AUROC and AUPRC of your models.
3. The current Introduction is a bit of too long, and several descriptions seems more suitable to put in the Discussion section.

Reviewer #2

(Remarks to the Author)

What are the major claims of the paper?

This manuscript utilizes a machine learning approach to predict dementia risk in non-caucasian populations (Hispanic Latino Americans and African Americans) and compares this to traditional polygenic risk models as well as APOE allele dosage models. The utilized two independent cohorts for this study, UCLA Health for training and All of Us for validation. They refined SNPs for their predictive models using functional genomic information and identified ancestry specific risk genes and pathways.

The authors claim that their elastic net model for dementia risk prediction has superior performance compared to traditional PRS and APOE allele dosage models.

Are they novel and will they be of interest to others in the community and the wider field?

In my opinion this manuscript will be of interest to the genetics, dementia and personalised medicine community as it addresses a number of pressing research questions. The authors cover the topic of polygenic risk in non-caucasian populations which is a well established research gap. They also address the use of machine learning approaches in polygenic risk prediction and benchmark their elastic net model against traditional PRS methods. Lastly the phenotype they explore is dementia which is an increasing public health concern due to an aging population.

If the conclusions are not original, it would be helpful if you could provide relevant references.

The conclusions are original and well founded based on the results of this study.

Is the work convincing, and if not, what further evidence would be required to strengthen the conclusions?

The work is convincing although somewhat underpowered. Demonstrating the improved performance of their elastic net in a larger sample size of HLA and AA populations will support their claim further.

On a more subjective note, do you feel that the paper will influence thinking in the field

Yes this adds to a growing evidence base for need to use population-specific polygenic risk models in complex genetic diseases such a dementia. It also supports the use of machine learning approaches for stronger predictive performance which is gaining traction in the field of polygenic risk. With growing cohort sizes and more readily available compute (e.g. cloud computing) machine learning approaches will become more prolific and relevant to the field.

We would also be grateful if you could comment on the appropriateness and validity of any statistical analysis,

The study is robust and follows a thorough process for data QC and statistical/computational analysis. It is worth noting that the datasets are quite imbalanced between cases and controls, although the authors do acknowledge this and use AUPRC as well as AUC to measure model performance, which somewhat mitigates for this imbalance. However the imbalance may still influence the reliability of their machine learning models. Likewise, the cohort sizes are quite small in both training (UCLA – 123 HLA and 84 AA cases) and validation (All of Us – 81 HLA and 181 AA cases) cohorts which will impact the power of the predictive performance for elastic net models. This is a known issue in studying underrepresented populations

Ability of a researcher to reproduce the work, given the level of detail provided.

The authors share their code for analysis on GitHub and GWAS summary stats are publically available. UCLA and All of Us cohorts have restricted access so anyone wishing to reproduce the work will need to go through the panel review process.

Author Rebuttal letter:

Reviewer #1 (Remarks to the Author):

Comments for the Authors

Fu and his/her colleagues explored how genetic risk modelling provide additional benefit to dementia. By integrating SNPs identified from pre-GWAS analysis, authors leveraged Elastic Net to develop prediction models and compare with APOE and PRS to demonstrate the additive values across genetic ancestry groups. The study design and objective are straight forward and easy to follow. The study cohort of UCLA EHR data (discover cohort) and All of US (external validation cohort) are relative large-scaled enough to support the findings. Overall, I think this is an interesting and necessary study; however, and I have several suggestions pertaining to the methodology that hope to improve the study.

Major comments:

1. Why setting case-control ratio in this study as 1:3? Your results of Precision, recall, specificity and accuracy that derived from 2-by-2 contingency matrix were highly depend on the ratio of

cases versus controls. In such 1:1, 1:2, ..., 1:5 would largely impact your results. Since your data is a community-based cohort, and it has much more controls, why not put all of them in your study?

We appreciate the reviewer's concern regarding the case-control ratio and its potential impact on our study's results. We agree that the prevalence of dementia in our study population can affect accuracy and precision, in addition to sensitivity and specificity in practice (Altman and Bland, 1994; Murad et al., 2023).

In response to the reviewer's suggestion, we have revised our study cohort to include all eligible cases and controls that met our initial inclusion criteria. The Hispanic Latino American (HLA) sample included 610 patients with 126 dementia cases, and the African American (AA) sample included 440 patients with 84 dementia cases. We continue to use AUPRC as our primary evaluation metric, as it provides a more balanced view of model performance with low prevalence conditions (Davis et al., 2006).

Table 2 presents the updated model performances. In summary, our proposed Elastic Net SNP (SNPs from AD + Neuro GWASs) models still demonstrate an overall improvement in dementia prediction across both ancestry groups. We have updated the manuscript's Methods, Results, and Discussion sections accordingly.

References:

Altman, D. G., & Bland, J. M. (1994). Statistics Notes: Diagnostic tests 1: sensitivity and specificity. *BMJ*, 308(6943), 1552.

Davis J, Goadrich M. The relationship between Precision-Recall and ROC curves. In: Proceedings of the 23rd International Conference on Machine Learning - ICML '06. ACM Press; 2006:233-240. doi:10.1145/1143844.1143874

Murad, M. H., Lin, L., Chu, H., Hasan, B., Alsibai, R. A., Abbas, A. S., Mustafa, R. A., & Wang, Z. (2023). The association of sensitivity and specificity with disease prevalence: analysis of 6909 studies of diagnostic test accuracy. *CMAJ : Canadian Medical Association journal*, 195(27), E925–E931. <https://doi.org/10.1503/cmaj.221802>

2. Elastic Net is actually a mixture of lasso and ridge, two regularized methods on top of regressions, please specific how did you determine the hyperparameter alpha of the proportion between L1 and L2 regularizations. How many SNPs remained in the final model after regularization?

3. Could you please provide more explanation about why you chose the Elastic Net, rather than other models to identify the variables that are associated with different dementia statutes, it would be nice to provide more comparable results of other models.

We thank the reviewer for pointing out these two questions regarding our use of Elastic Net regularization. We selected Elastic Net because it is particularly effective in situations with numerous predictor variables, such as single nucleotide polymorphisms (SNPs), and in addressing potential multicollinearity among these predictors.

To clarify, we have included the following text in our Methods section (Lines 197-203), which provides a brief introduction to Elastic Net and our method for determining the hyperparameter α :

“The (4) model involved the application of Elastic Net regularization, which combines the benefits of both Lasso (L1) and Ridge (L2) regression methods to enhance model stability and variance handling. This technique aids in variable selection by reducing the coefficients of less relevant variables to zero, simplifying the model, and improving its ability to manage multicollinearity (Zou and Hastie, 2005). The hyperparameter α , which balances L1 and L2 regularization, was optimized using a grid search to maximize the penalized likelihood within each training set.”

The number of SNPs retained in the final models after regularization for each ancestry group is mentioned in the Results section (Lines 319-321):

“In our analysis of the best-performing Elastic Net SNPs models, we examined the features selected by each model. According to results from bootstrapping (at least 95% of the 1,000 iterations), the HLA and AA models identified 28 and 31 risk SNPs, respectively.”

The method for selecting SNPs included in the final model is detailed in the Methods section. This includes an explanation of cross-validation and bootstrapping, which will be elaborated on in response to Q4 and Q5 (see below).

Additionally, in response to the reviewer's suggestion, we have included two more models,

gradient boosting machine (GBM) and XGBoost, to compare different machine learning approaches. Similarly, we optimized the hyperparameters for each model using a grid search approach within each training set. Both models, however, did not perform as well as the linear Elastic Net SNP models, as presented in Table 2 and Supplementary Figure 2.

References:

Zou H, Hastie T. Regularization and Variable Selection via the Elastic Net. *Journal of the Royal Statistical Society Series B (Statistical Methodology)*. 2005;67(2):301-320.

4. The authors employed a 5-fold cross-validation methodology, how it performed? Did you tune your parameters using the CV scheme? If so, I believe your model might be overfitting as your testing data was used for selection of SNPs. You can only optimize your model using the training data, while the testing need to held out and merely used for evaluation. Please be careful and clearly state this part.

5. The cross-validation and 1000 permutation confused me. CV means you train and evaluate your model separately and iteratively, how this combined with permutation? Did you combined predicted probabilities from each testing fold of data and then evaluated through permutation? I believe a supplementary figure of flowchart could better to illustrate your training and evaluation procedures.

6. The permutation seems exact the bootstrap method, and your p-value was based on a hypothesis that the metrics used are simple sums of IID random variables. This is not a trivial problem as the hypothesis not stand for such situation, and comparing them additionally requires taking into account the correlation between metrics calculated on the same sample. Please consider DeLong statistics for AUROC.

We appreciate the reviewer's insightful questions regarding cross-validation and permutations in our study. In response, we have clarified the 5-fold cross-validation procedure and revised our permutation methods to a bootstrapping methodology. The bootstrapping approach involves random sampling with replacement for all participants in each genetic ancestry group. Additionally, we included a figure (see below) to illustrate the 5-fold cross-validation process. The Methods section now includes the following details (Lines 210-215):

"We employed a 5-fold cross-validation methodology across all models to evaluate performance, with final results reported on the combined hold-out testing sets (Figure 1). To enhance the robustness of our findings, we utilized bootstrapping (Efron et al., 1994) to determine feature importance, determine confidence intervals (CIs), and establish statistical significance. Specifically, we repeated the modeling process 1,000 times using random sampling with replacement of all subjects (cases and controls) within the analytical sample set of each GIA group."

During the modeling phase, parameter tuning was performed within each training set during cross-validation, and model performance was reported only on the combined hold-out test set. This is the standard cross-validation approach and minimizes overfitting. For each bootstrapping iteration, we sampled the entire modeling population (cases and controls) with replacement, followed by the modeling-evaluation step. Confidence intervals (CIs), statistical significance, and final selected features (present in at least 95% of the 1,000 iterations) were determined using this bootstrapping strategy.

We acknowledge that our data may not consist of independent and identically distributed random variables, which can limit the precision of p-value estimates when using bootstrapping tests on correlated metrics from the same sample. To address this, we revised our analyses using DeLong's test (DeLong et al., 1988), which is suitable for comparing two AUROC values derived from identical observations. Since there is no equivalent test for AUPRC comparisons, we employed the paired Wilcoxon signed-rank test (Conover W., 1999) to compare AUPRC using the bootstrapping results. We have updated the manuscript and relevant tables (Table 2, Supplementary Figure 2, Supplementary Table 3) to reflect these changes.

References:

Conover W. *Practical Nonparametric Statistics* (3rd Ed.). John Wiley & Sons, Inc.; 1999.

DeLong ER, DeLong DM, Clarke-Pearson DL. Comparing the Areas under Two or More Correlated Receiver Operating Characteristic Curves: A Nonparametric Approach. *Biometrics*. 1988;44(3):837-845. doi:10.2307/2531595

Efron B, Tibshirani RJ. *An Introduction to the Bootstrap*. CRC Press; 1994.

7. The authors defined dementia, AD, VD, etc based on ICD-10, I am curious about what the diagnostic criteria for these disease? Are they following the same criteria of UCLA and All of US cohort?

The International Classification of Diseases (ICD) codes are a system of alphanumeric codes used by healthcare providers to classify and code all diagnoses, symptoms, and procedures recorded during hospital care (World Health Organization, 2004). These codes are entered by

clinical providers during patient visits, adhering to similar general diagnostic criteria. Although the clinical providers at UCLA and those in the All of Us cohort are different, both groups of clinicians use these general diagnostic criteria to enter ICD codes for their patients. The ICD-10 criteria offer a standardized language for reporting and monitoring diseases like Alzheimer's Disease and Vascular Dementia across various clinical settings. For instance, ICD-10 classifies different forms of Alzheimer's Disease by considering factors such as the age at onset and specific symptoms like memory loss, cognitive impairment, and behavioral changes. Alzheimer's Disease is coded as G30, with subcategories detailing onset and symptom progression. Vascular Dementia is coded as F01, emphasizing its vascular origins. The diagnostic criteria for Vascular Dementia include evidence of cerebrovascular disease linked to cognitive decline that significantly disrupts daily activities.

While detailed clinical guidelines exist for these diagnoses (McKhann et al., 1984; Engelhardt et al., 2011), clinical providers may enter ICD codes for these diagnoses without meeting all clinical diagnostic criteria. Therefore, one limitation of using electronic health records data is that the precision of disease diagnoses based on ICD codes may vary compared to a gold standard of research criteria or autopsy findings. We have acknowledged this limitation in the Discussion section (Lines 519-523).

"Finally, although detailed clinical guidelines for disease diagnoses exist, 77,78 clinical providers may adapt these criteria to fit specific research focuses or populations. This adaptation can lead to variations in diagnostic criteria across different studies or clinical practices. Consequently, the precision of dementia diagnoses based on ICD-10 codes may vary compared to a gold standard of research criteria or autopsy findings."

References:

- Engelhardt, Elias, Carla Tocquer, Charles André, Denise Madeira Moreira, Ivan Hideyo Okamoto, and José Luiz de Sá Cavalcanti. 2011. "Vascular Dementia: Diagnostic Criteria and Supplementary Exams. Recommendations of the Scientific Department of Cognitive Neurology and Aging of the Brazilian Academy of Neurology. Part I." *Dementia & Neuropsychologia* 5 (4): 251–63. <https://doi.org/10.1590/S1980-57642011DN05040003>.
- McKhann, G., D. Drachman, M. Folstein, R. Katzman, D. Price, and E. M. Stadlan. 1984. "Clinical Diagnosis of Alzheimer's Disease: Report of the NINCDS-ADRDA Work Group under the Auspices of Department of Health and Human Services Task Force on Alzheimer's Disease." *Neurology* 34 (7): 939–44. <https://doi.org/10.1212/wnl.34.7.939>.
- World Health Organization. 2004. ICD-10 : international statistical classification of diseases and related health problems : tenth revision. World Health Organization. <https://iris.who.int/handle/10665/42980>.

Minor comments:

1. Please add the number of participants, target cases to your abstract.
2. Also, please add the range of AUROC and AUPRC of your models.
3. The current Introduction is a bit of too long, and several descriptions seems more suitable to put in the Discussion section.

We also thank the reviewer for these minor comments. We have made adjustments in the manuscript accordingly.

Reviewer #2 (Remarks to the Author):

The work is convincing although somewhat underpowered. Demonstrating the improved performance of their elastic net in a larger sample size of HLA and AA populations will support their claim further.

The study is robust and follows a thorough process for data QC and statistical/computational analysis. It is worth noting that the datasets are quite imbalanced between cases and controls, although the authors do acknowledge this and use AUPRC as well as AUC to measure model performance, which somewhat mitigates for this imbalance. However the imbalance may still influence the reliability of their machine learning models. Likewise, the cohort sizes are quite small in both training (UCLA – 123 HLA and 84 AA cases) and validation (All of Us – 81 HLA and 181 AA cases) cohorts which will impact the power of the predictive performance for elastic net models. This is a known issue in studying underrepresented populations

We appreciate the reviewer's detailed comments. We acknowledge the issue of having an imbalanced dataset in our current study. To address this, we evaluated our model using the AUPRC and optimized the MCC, both of which are effective for imbalanced datasets. We also recognize that the prevalence of cases may impact our model's performance (see response to Reviewer #1, Q1).

In addition, we have added the small sample size as a limitation in the Discussion section (Lines 516-519).

“Thirdly, the limited number of dementia cases in our non-European GIA samples, after applying inclusion criteria, constrains the generalizability of our findings. Future studies should aim to replicate these findings in larger samples for each GIA to enhance their robustness.”

Version 1:

Reviewer comments:

Reviewer #1

(Remarks to the Author)

I thank the authors for their careful revision and clarification of all my concerns.
